# Happy Existentialist Metaphors: Merleau-Ponty's Flesh of the World and the Chandos Complex

## Annabelle Dufourcq

Faculty of Philosophy, Theology and Religious Studies, Radboud University, Houtlaan 4, 6525 XZ Nijmegen, The Netherlands; annabelle.dufourcq@ru.nl

**Abstract:** This article investigates the meaning of Merleau-Ponty's concept of the flesh of the world. This concept brings a cosmological tone to existentialist phenomenology and challenges the grim and gnostic approach that prevails in Heidegger's and Sartre's works in particular. Is horror the key mood in ontology as argued by Malabou? This article contends that bright metaphors and magic realism are at least as fundamental, but under one condition: ontology must come to terms with what the author has coined as the "Chandos complex", namely a form of ambivalence and oscillation between Gnosticism and holism that makes both positions fake and hollow. Dreaming of being one with the world and fantasizing an estrangement from nature work hand in hand and are equally staged. Merleau-Ponty's philosophy occasionally falls prey to the Chandos complex, which makes his concept of the flesh of the world vulnerable to criticism. This article examines the claim put forward by Renaud Barbaras that "the flesh of the world" is a failed metaphor. It argues that this blissful metaphor is ontologically fundamental as soon as its intrinsic paradoxes are recognized and accepted: the Chandos complex then becomes the key to an ontology that recognizes the imaginary as an essential dimension of being. At stake is an essential link between ontology on the one hand and, on the other hand, metaphors as well as myth-building and narrative-building processes.

**Keywords:** phenomenology; ontology; realism; holism; gnosticism; magical realism; imagination; emotion; narrative; metaphor

## 1. Introduction

Existentialism famously overturns the classic link between imagination and nothingness into an ontological superpower, or rather superweakness. Hence, the mood that is the most attuned to the ontological perspective is classically defined as coming into play when the confidence in a solid world crumbles. Contingency eats away at things around us so that they deliquesce into grinning and grimacing faces and turn into a creepy pasteboard decor. Consider the viscous in Sartre's *Nausea* or the nocturnal "there is" in Levinas' *Existence and existents* ([Levinas 1988](), pp. 54–55): the fantastic crops up when the "dissociation between existing and existents" becomes manifest in the world, through the deconstruction of the real, as Catherine Malabou argues in "Pierre Loves Horranges" ([Malabou 2013](), pp. 104–5). Thus, according to Malabou, the fantastic in philosophy wears the face of horror. Yet, the picture would have been substantially different if Malabou had drawn her inspiration from Merleau-Ponty, Bachelard, or even Husserl, instead of Heidegger, Levinas and Sartre. Merleau-Ponty takes issue with the gnostic tendency that animates Heidegger's ontology ([Merleau-Ponty 1995](), p. 122).[1] The oblivion of Being, the *Grundstimmung* of *Angst* and, in Modern times, *Erschrecken* (terror) and the solitude of the Dasein tie us to an irreducible dimension of estrangement (*das Unzuhause*) that the philosopher cannot but approach in an insincere, Janus-faced manner: partly the complacent know-it-all, partly the distressed prophet of doom. By contrast, Merleau-Ponty—who devoted his master's thesis research to Plotinus' philosophy—resolutely opts for more radiant images. He embraces a fundamental mood that Bachelard described in *The Poetics of Space* as the "enthusiasm" that "is born

of adherence to the felicity of an image", by way of an alternative to the propension of existentialism to showcase images of man as "cast [*jeté*] into the world" (Bachelard 1994, p. 7). Indeed, Bachelard argues, since no being "cast out" would occur without a more original "being of within" (Bachelard 1994, p. 7), phenomenology should start from images of happiness and the "happiness of image [*un bonheur d'image*]" (Bachelard 1994, p. 9). However, existentialism is never without vulnerability and inner wrenching. The images of the flesh of the world in Merleau-Ponty's philosophy are far from tension-free: things "bleed" (Merleau-Ponty 2002, p. 53), the flesh of the world grows monstrous pseudopods that harm each other and it sometimes ossifies and represses itself. Moreover, the very idea of the "flesh of the world" may hinge on a failed metaphor precisely because a gap must remain between the living flesh and the matter of inert things, as Renaud Barbaras (2011) argued. Hence, the daunting challenge: can existentialism avoid the gnostic horror? Is the concept/metaphor of the flesh of the world too tender as Deleuze claimed? Does its radiance cover a poisonous heart? Should we consequently dispose of it? By framing the question in such terms, I want to point out that ontology may be a play with what Queneau called "little portable cosmogonies"; it may be a way of telling oneself stories—in the sense of creating narratives, with a measure of dishonesty and self-deception when these images are not recognized as such and are dramatized and jargonized. In order to challenge this latter tendency, I will argue that, at work in the existentialist ontology and still in Merleau-Ponty's ontology, is what I have called elsewhere the Chandos complex, in reference to Hofmannsthal's *Brief des Lord Chandos an Francis Bacon*, a letter in which Lord Chandos, a fictitious doppelganger of Hofmannsthal, sadly pronounces the divorce between the human language and the overwhelming intuition of fusion with the universe, while describing this intuition with a host of picturesque details and vivid images. However, let me be clear: showing the essential link between the existentialist ontology and the imaginary is certainly a way of deflating ontology, as it were, but not of reducing it to psychology. It is precisely at the level of the flesh of the world that being and not only the human being, tells itself stories. I will contend that the irony and the ambivalence of Chandos, as manifested in a fanciful and playful piece of literature, stand at the heart of ontology in its most fundamental form.

## 2. The Flesh of the World

The concept of the flesh of the world is undoubtedly introduced by Merleau-Ponty in an anti-gnostic move to overcome the idea of a fundamental rift between humans and the world.

The human mind may seem to possess the exceptional ability to thematize objects as such (Heidegger 1995, p. 248; Merleau-Ponty 1960, pp. 114–24) and allow them to appear—namely, to become free-floating *eidola*. It can then be tempting to regard human consciousness as a "hole in being" (Merleau-Ponty [1942] 1990, p. 136)[2] and to define existence as nothingness and, thus, as the power of spacing and cutting. Hence, an uncanny revival of Gnosticism in Heidegger and Sartre for instance: the human is outlined as a stranger to this world, an alien doomed to nostalgy and solitude (Heidegger 1995, §2) and whose existence *as* self-distance is the condition without which beings would never appear.

However, Merleau-Ponty argues, the human cannot be an ontological enclave of existence in being: no encounter *with* a real, transcendent world would ensue from such a radically detached being. Perception must be prepared and motivated from within the perceptible objects; it must stem from an intrinsically phenomenal world. The lineaments and the diversity of the perceived world are suggested to the perceiving subject by the things themselves. Without a phenomenologizing process that arises from the objects, *nothing* would ever be thought. The perceiving subject is therefore essentially a body *in* the world, *part of* the world and its nature as a perceiving and feeling body cannot be an exception. In other words, my flesh is a certain concretion of what must be defined as the *flesh* of the world and this, Merleau-Ponty adds, "is no analogy or vague comparison and must be taken literally" (Merleau-Ponty 1968, p. 133).

Indeed, lights, shadows and emerging consistent lines of style in the flux of adumbrations guide the exploratory movements of my body. Things play *melodies*[3] that resonate in my flesh and that my body so to say takes up: thus, for instance each color, even through a subliminal stimulation, has a way of resonating, a rhythm and a style that make it transposable into a certain emotional state of my body, namely into postures, states of muscular tension and micro-movements, which, in their turn are transposable into a set of sound sensations (Merleau-Ponty 2012, pp. 216–23).

The key here is that Merleau-Ponty defines sensory qualities by a certain structure, or *Gestalt* and as diacritical beings—namely as beings that emerge and define themselves (κρίνειν: to separate, discern, order, arrange) through (διά) a flux of adumbrations—instead of regarding them as ultimately irreducible intuitive contents, raw data, undifferentiated, instantaneous and punctual "jolts [chocs]" (Merleau-Ponty [1945] 2001, p. 9; 2012, p. 3). As a result, a dance and a song can be two variations on the same structure. The same goes for a perceptive appearance and a painting for instance. These melodies embedded in the ways of being of qualities, things and living bodies can be played with many surprising variations; hence, the possibility of original and still accurate metaphors.

This complicity, the dialogue between melodies that solicit and respond to each other is the homogeneity that binds together my flesh and the flesh of the world. Merleau-Ponty thus insists on referring to a literal identity between the stuff [*étoffe*] the human flesh is made of and the stuff all other entities in this world are made of: "Visible-seer [Voyant-visible] ( . . . ) They both must be abstracts from one sole tissue [abstraits d'une seule étoffe]" (Merleau-Ponty 1968, p. 262; [1964] 2001, p. 310),[4] and in *Eye and Mind*: "the world is made of the very stuff of the body [le monde est fait de l'étoffe même du corps]" (Merleau-Ponty 1961, p. 19; 1993, p. 125).

### 3. A Failed Metaphor?

However, something does not add up in this theory of the flesh of the world and Merleau-Ponty's argument is in fact shaky at the core, so that Merleau-Ponty's claim to literality may be the very symptom of an adventuresome metaphor, worst: of a deficient metaphor.

In *Les trois sens de la chair. Sur une impasse de l'ontologie de Merleau-Ponty*, Renaud Barbaras (2011) expertly highlights the flaws that undermine Merleau-Ponty's reasoning. A discrepancy remains and cannot but remain between my flesh and the flesh of the world. In fact, Merleau-Ponty continues to emphasize that there is a specificity of the human flesh and his reasoning hinges on this idiosyncrasy of the human flesh. Indeed, the whole reasoning rests upon the dissonance, in the first instance, between a set of characteristics that are normally ascribed to the human body (sentient, narcissistic, desiring, the source of symbolism . . . )[5] and the absence of these characteristics in the traditional definition of inert bodies. Merleau-Ponty thus insists: with the concept of the flesh of the world, our body is taken as "measurement, standard [*étalon*] of the world" (Merleau-Ponty 2003, pp. 224–45). At stake is "an *extension* of the narcissism of the body" (Merleau-Ponty 2003, p. 225, my emphasis). There is something like a *becoming*-flesh of the world: "The thickness of the body means I have to go unto the heart of the things, by making myself a world and by making them flesh [*en me faisant monde et en les faisant chair*]. ( . . . ) Caught up in the tissue of the things, [the body] draws it entirely to itself, incorporates it, and, with the same movement, communicates to the things upon which it closes over that identity without superposition, that difference without contradiction, that divergence between the within and the without that constitutes *its* natal secret" (Merleau-Ponty 1968, pp. 135–36, my emphasis).

If the flesh is specifically the flesh of my human body, how can it be the flesh of the world? Yet, if the flesh of my body is homogeneous to the matter of the world, the transposition is possible and its ontological power is effective... but its explanatory power vanishes, for there is no specific characteristic that can be transposed in an illuminating manner from my flesh to the flesh of the world.

Consequently, Barbaras claims, if the transposition is possible, it is only the body as a *Körper* that can become the model for the matter of the world, or, put another way: the transposition is not really a transposition, it tells us nothing that we did not know already. As for the concept "flesh of the world", it is, in Barbaras' view, the result of an illegitimate transposition: the rabbit that is pulled out of the hat (the flesh of the world) is not the same as the one that was put in it (the matter of inert things, the body as a *Körper*) (Barbaras 2011, pp. 16–17). The *meta-phor* (from the Greek "transfer") failed. This metaphor actually remained a fission and it was necessary to supplement the latter with a substitution. When Merleau-Ponty calls this a *metamorphosis,* he therefore actually performs a cheap magic trick. A working note somehow finally lets the rabbit out of the bag: "*The* flesh of the world is not *self-sensing [se sentir] as* is my flesh—It is sensible and not sentient—I call it flesh, nonetheless . . . " (Merleau-Ponty 1968, p. 250; [1964] 2001, p. 298).

I will argue that this criticism does not exhaust the Merleau-pontian concept of the flesh of the world: between the human flesh and the meaningful matter of things in the world, a relation of essential co-institution is at work, rather than a simple substantial identity. However, the fact remains that Merleau-Ponty puts the emphasis ambiguously and partly misleadingly on the "same stuff" the body and the world are made of. This dream of unity is, I contend, correlative to the fact that, more generally, Merleau-Ponty has not fully emancipated himself from the Gnostic conflictual cosmogony.

## 4. Merleau-Ponty's Oscillations

The ambivalence in Merleau-Ponty's philosophy takes the form of the coexistence of two opposing tendencies throughout his later works. The tricky task of integrating them into a coherent philosophy is left to the reader.

On the one hand, Merleau-Ponty regularly emphasizes the unity and the eternal truth of the flesh as a non-anthropological concept (Merleau-Ponty 1968, p. 136; [1964] 2001, p. 177). As Deleuze highlighted it, the flesh is first and foremost a figure of unification: "flesh of the world and flesh of the body that are exchanged as correlates, ideal coincidence" (Deleuze and Guattari 1994, pp. 178–79). To be sure, the flesh, insofar as it is the name of a thick, embodied and perspectival meaning, has nothing in common with the transcendental subject as the ultimate source of a perfect intuition, absolute unity and full clarity. However, the theory of the flesh still hinges on the model of an ultimate sympathy and even homogeneity (Merleau-Ponty 1968, p. 114; [1964] 2001, p. 150. See also "harmonie préétablie" (Merleau-Ponty [1964] 2001, p. 173) between the subject and the world. "We are within [*nous sommes intérieurs à*] life, within the human being and within Being and ( . . . ) [Being] is in us as well ( . . . ) This environment [*ce milieu*] of brute existence and essence is not something mysterious: we never quit it [*nous n'en sortons pas*], we have no other environment [*nous n'en avons pas d'autre*]." (Merleau-Ponty 1968, pp. 116–17; [1964] 2001, p. 154). "The sensible, Nature, transcend the past present distinction, realize from within [*par le dedans*] a passage from one into the other Existential eternity. ( . . . ) Do a psychoanalysis of Nature: it is the flesh, the mother." (Merleau-Ponty 1968, p. 267; [1964] 2001, p. 315). There is an "impossibility of meaninglessness" (Merleau-Ponty 1968, p. 117) and in a way "all is true" (Merleau-Ponty 1998, p. 37).

These assertions conflict with passages in which Merleau-Ponty arraigns objective thought and warns us against a "cultural regimen in which there is neither truth nor falsehood concerning humanity and history, into a sleep, or nightmare *from which there is no awakening*" (Merleau-Ponty 1993, p. 122). What rupture, what oblivion could ever happen if the origin is a wisdom devoid of exteriority? How could we lose sight—worst: lose to the extent that *no awakening is possible*—of what according to Merleau-Ponty is always at work in us, as the core of our very being? It is one of two things. Either the flesh is universal and all-pervading and we can never lose it *in any way*; or it was never really us and it was certainly never this unity that Merleau-Ponty has described numerous times. If we haven't lost anything, what *difference* can Merleau-Ponty hope to make with his theory of the flesh of the world and how can such a theory can even *arise* and *puzzle* us? As Merleau-Ponty



dramatizes the accusation against "the objective thought" and correlatively "idyllizes", if you will pardon the expression, the salvation by the Flesh, the problem builds up into an aporia.

This conflict is indeed the same as the one described by Barbaras between the flesh in the ontic meaning of the term (my body), the flesh as transcendental or the human ability to turn the sensible into vision and the flesh of the world as the meaningfulness of a "sensible but not sentient" Being. On this issue, again, Merleau-Ponty oscillates by using alternately the vocabulary of identity (same stuff, same magma, Merleau-Ponty 1996, p. 211) and the vocabulary of difference ("sensible and not sentient", "if the body is a thing among things, it is so in a stronger and deeper sense than they" Merleau-Ponty 1968, p. 137). The same oscillation returns when Merleau-Ponty decisively claims that the concept of the flesh of the world *is not* a vague comparison (Merleau-Ponty 1968, pp. 221–22; 1996, p. 211), and then, a few pages later, proposes to provide more metaphors, as he is trying to describe the unity of the nevertheless twofold body ("the body sensed and the body sentient") and must venture several mutually rectifying descriptions: "if one wants metaphors, it would be better to say . . . " (Merleau-Ponty 1968, pp. 133, 138).

The problem with ambivalence is that it ends up doing the opposite of what it says and cannot but say the opposite of what it does. On the one hand, it manages to play both sides of the coin, but, on the other hand, ontologically speaking, it fails to overcome the traditional dualities: it continues to hope for a conciliation of *the opposites regarded and maintained as such*. This is precisely what I have called the Chandos complex.

## 5. The Chandos Complex

In 1603, Lord Chandos, "younger son of the Earl of Bath, wrote to Francis Bacon, later Baron Verulam, Viscount St. Albans, apologizing for his complete abandonment of literary activity" (von Hofmannsthal 2008, p. 69). Chandos at one time was a prolific author in a lyrical and idealist vein. Yet, under the blows of the most radical existential crisis, the radiant spiritual unity that was pervading every being has vanished. "For me everything disintegrated into parts, those parts again into parts; no longer would anything let itself be encompassed by one idea. Single words floated round me; ( . . . ) whirlpools which gave me vertigo and, reeling incessantly, led into the void" (von Hofmannsthal 2008, p. 74).

Yet, this text is also a jest—or at least I cannot resist the pleasure of reading it that way. It was published in 1902 by Hugo van Hofmannsthal, who playfully creates an effect of reality—Bacon is a historical figure, Chandos is a fiction—and of autofiction. Chandos is the epitome of the romantic ideal that was Hofmannsthal's first love. In that, Chandos is certainly Hofmannsthal's doppelganger. However, the paradox embodied by Chandos culminates when it appears that Hofmannsthal has by no means given up writing, as evidenced by this malicious letter in which the author explains the renunciation of words with ample holistic lively descriptions and abundant and eloquent lamentations.

The reference to Bacon is well suited for the deconstruction of a metaphysical and illusory language ("I experienced an inexplicable discomfort for so much as uttering the words 'mind', 'soul' or 'body'", von Hofmannsthal 2008, p. 73, modified translation) and a pragmatic call to go back to experience. However, what was accomplished with optimism and as a salutary return to empirical reality by Bacon becomes with Chandos—in this sense a bad student (Le Rider 1994) or perhaps simply a man more deeply in contact with the agony of Modernity—a crisis and the source of aporias and despair.

The letter of Lord Chandos thus embodies ambivalence at its height, to a point verging on sublimity and farce. Chandos oscillates between moments of rapture and holistic intuition and moments of confusion and fragmentation.

Several commentators have highlighted the similarities between Chandos' ideas and the key features of Existentialism in Sartre's work especially (Evelton 2009; Huemer 2003). In the *Chandos Letter* as well as in Sartre's *Nausea* the real resurfaces as a raw presence, the absurd fact of existence and a silent mystery. The gap between subject and reality is manifested by a diffuse anxiety and an impression of emptiness and meaninglessness.

However, Sartre also sensed this paradox as he eventually confesses in *Les Mots:* "I gaily demonstrated that man is impossible; Fake to the marrow of my bones and hoodwinked, I joyfully wrote about our unhappy state" (Sartre 1965, p. 252).

Likewise, as perfectly exemplified by *Don Quixote*, while chivalric romance thematized the unity between the ideal and the real, the modern novel claims to return to the real itself, in the guise of the raw, resistant, brutal and inhuman real. Yet, this realist turn is awkwardly accomplished through fiction, which is also a manner of giving the game away. By hardening the conflict between the subject and the world, modernity creates the correlative myths of a separate subject and a, so to say, really real reality. The latter is fantasized and treasured as—paradoxically—the only reality worthy of the name and totally foreign and unspeakable. Nevertheless, this separation is entirely staged and, in the end, all this holds only if one constantly does and experiences the opposite of what one says.

I define the Chandos complex as two opposing models that repel each other and yet necessarily support each other. On the one hand, without the yearning for unity, the experience of the sensible flux would never give rise to any impression of fragmentation, frustration, unease, or loss of meaning. Without a minimal relevance of language, Chandos' despair in the face of the ineffable would never bear such luxuriant descriptions. On the other hand, without fragments and individualizing processes at work in the organic sensible flow, Being would be reduced to a mute and perfectly self-coincident totality: there would not be any *reality to experience or even to paradoxically miss.* No mediation, no drama, no effect of estrangement. Chandos in a regime of innocent bliss would write nothing.

This complex was somehow exposed by Bergson or Merleau-Ponty when they described idealism and realism as secret accomplices, but it proves to be still alive in existentialism and beyond, including, to a certain extent, in Merleau-Ponty's texts, as I have argued above, as well as, for that matter, in Deleuze's reference to a pure "Outside [*Dehors*]" (Deleuze and Guattari 1994, p. 59). Moreover, Hofmannsthal's letter casts a new light on this "contradiction", precisely by playing with it. Planning to *overcome* the complex would actually be a way of succumbing to it again. It then becomes possible to *have it all*, as it were, at the expense—or with the gain—of a pinch of irony. Through the lens of a literary and fanciful approach, the paradox can become a meaningful and illuminating genetic ontology. From Chandos' perspective and from the perspective of those who fully commit to his crisis, the letter tells a story of failure and despair. Yet, the distance introduced by Hofmannsthal through exaggerations and dramatizations helps us understand the mechanisms of this crisis and reframe them in a new holistic-diacritical perspective. Hence, my choice of the term "complex", borrowed from Bachelard's philosophy of the imaginary. The Chandos complex consists in a certain dynamic that orients and gives momentum to our imaginative forces and to our existential relation to the world, but that can also be taken up and fine-tuned by the imagining subject in a more active fashion. I can lament or laugh with Chandos. I can also encounter major ontological concepts that manifest under the guise of the dramatized poles that structure Chandos' adventures (fragmented sensations, the void, the whole, the all-pervading spirituality, the isolated subject, the real . . . ) and I may discover my unbreakable connection to the meaning of every reality even at the reverse side of the frustration that I experience empathizing with him.

As a result, I contend that the solution to the contradiction that was highlighted by Barbaras is not to dispose of the Merleau-Pontian concept of the flesh of the world. Let us turn again to Merleau-Ponty's ontology in the light of our analysis of the Chandos complex.

## 6. The Ontological Imaginary Dimension of the Flesh

Merleau-Ponty does not exclusively operate in the ambivalent mode. He also provides us with the means to reframe his oscillations into a more complex dynamic.

First, in several places, Merleau-Ponty amends his claim that my flesh and the flesh of the world are one and the same. "There are two circles, or two vortexes, or two spheres, concentric when I live naïvely, and as soon as I question myself, the one slightly decentered

with respect to the other" (Merleau-Ponty 1968, p. 138). Second, and correlatively, Merleau-Ponty notes, flesh "is not matter, is not mind, is not substance", "it is an element of Being" (Merleau-Ponty 1968, p. 139). It is important to point out that the flesh is only "an" element *among others*. The difference of my flesh is thus acknowledged, but what makes this difference ontologically crucial is not essentially that my body would allegedly bring the ability to sense as its specific property.

Like the presocratic elements, my body can lend its style of being to other beings so that they can transpose themselves in it and express their own style of being through it. In elemental ontologies, everything is somehow fire, air, water or earth. Merleau-Ponty revives this approach when he states that, as it were, everything is flesh. This transposition happens for instance when I turn a melody into a dance, or a color into a certain state of muscular tension and, simultaneously, in a sound and a taste. My body *as an element* is nothing like a *Körper*, or a clearly circumscribed part of the world with specific *properties*. It is always already a concretion that polarizes and crystallizes a more fundamental "imaginary body" (Merleau-Ponty 1968, p. 262).[6] In addition, this imaginary body always exceeds the objective body and constitutes its halo or the lining that silently weaves and unweaves this objective body. It is this imaginary of the body, rather than the objective body, that is the "institution (*Stiftung*) of Being" according to Merleau-Ponty (1968, p. 262). This imaginary is part of the more general "imaginary texture of the real" (Merleau-Ponty 1993, p. 126), which consists in the flows of facets, sensations and events through which things emerge but that always exceed the boundaries of recognizable identities. In everyday perception, social conventions, survival strategies and for convenience purposes, we commonly ascribe to things and persons a certain recognizable set of appearances and modes of manifestation. However, the imaginary texture of beings is their ability to appear—and be—through an infinity of surprising new guises, including hybrid forms like synesthesiae, the strange and elusive line between land and sea described by Proust (1988, pp. 192–93), or the metamorphoses of landscapes through the works of art that explore them. Likewise, my body as "I, myself", remains vaguely circumscribed, since the boundary between anonymous states and states that are recognized as "mine" is porous. I build myself by appropriating and actively taking up sets of sensations and impulses that were born before "me", in "my" muscles, "my" unconscious sensible exploration of my environment and, even further, in a mode of being communicated to the living body by a color or an atmosphere. I suddenly find out, in hindsight: maybe this joy or this impatience defines "me", indeed, but, right before I noticed this emotion, was it already me or still the world? The imaginary of the body is neither the *Körper*, nor even my flesh in the sense of the body that I am. It is a swarm of powers, rhythms, desires, practical solicitations, introjections and projections on which I feed and through which I built myself. These textures, attitudes and habits borrowed from others, these words, these objects have enriched my body, namely the way I experience and represent it—the body image—but also, inseparably, its style, its metabolism and the way it forms and produces blood, tissues, fat, rhythms, postures, illnesses and skills.

The imaginary body is the institution of being in that there is indeed a fluid transposition of styles of being or *diacritical melodies* through various beings, including my body precisely at the level of the open flux of floating adumbrations. Does my body bring something special? Yes. Is my body the heterogenous exceptional being which turns everything into sensations? No. It emerges from scattered sensations that can be ascribed neither to things nor to a clear-cut organism. However, it does possess *its way of* sensing, conceptualizing, circumscribing and transposing sensations into words, works of art, scientific and philosophical theories.

My flesh and the flesh of the world resonate with each other. Therefore, they can't coincide. It is precisely the *differences* between styles of being, including, among others, the particularity of my body, that create the space for this resonance to operate. This diacritical being is also the condition without which a world, worthy of the name, would never exist, namely would not be a diversity of beings.

As a result, I contend that the only way to think the relation between my flesh and the flesh of things is to define it as a circular relation of mutual institution. Merleau-Ponty outlines the complex dynamic of this relation in *L'oeil et l'esprit* and *L'ontologie cartésienne et l'ontologie d'aujourd'hui*: a work of art and even the most prosaic vision are adventures that happen to Being. My vision or a work of art are "expressions"; they create something new and yet, they are born from the encounter with the world, namely, at a more fundamental ontological level, they continue or take up melodies that transcend them, that were born before them, resonate through them and will pursue their existence in other elements, although enriched by their metamorphosis through my body. The melodies have instituting—by contrast, with constituting—capacities, for they are open themes: they allow for contingent developments. Artists and, more generally, sensing bodies start from their contact with the world. Somehow, they continue what already is, but they also introduce a unique style as well as contingent inflections and variations. "The same thing is both out there in the world and here at the heart of vision—the same or, if you will, a similar thing, but according to an efficient similarity which is the parent, the genesis, the metamorphosis of being into its vision" (Merleau-Ponty 1993, p. 128). The world institutes my vision and my expressions, which, in their turn institute new variations on older themes, new facets of things or new things in the world and unique misadventures of Being. It is then impossible to do as though nothing happened: the contingent inflections become an integral part of Being (Merleau-Ponty 1996, pp. 203–4) and the *story* goes on. The new avatars of an institution could not be predicted from the previous instituting moments, but this is also why the instituted retrospectively institutes the instituting: after the fact, I recognize myself in this body, I was exploring the world with *my* eyes and *my* hands and it becomes necessary to say that the world was *meant to* be seen by my eyes, it was meant to be painted by Cézanne. The flesh of the world, the open imaginary texture of beings, demanded contingent developments. Thus, *in retrospect*, it demanded *these* developments. Contingence and necessity become one. I propose to use the term *co-institution*[7] to designate this ironical *destiny in hindsight*.

This dynamic and circular relation of co-institution between my flesh and the flesh of the world is precisely what makes this *metaphor* of the flesh ontologically relevant. It is the closest we can get to identity, when identity is actually dismissed as an illusion and an ontological dead end. Hence, Merleau-Ponty resorts, occasionally, to the approximation of this theory by formulas of identity, sameness, literalness and homogeneity. However, beyond the approximation, at stake is the fanciful story of being. This is where precisely Merleau-Ponty meets Chandos-Hofmannsthal and their cosmogony.

This is also where Merleau-Ponty's ontology of the flesh meets Bachelard's happiness of images. Bachelard's work was among the influences that played a crucial role in Merleau-Ponty's philosophy. Beyond Merleau-Ponty's explicit references to Bachelard's concept of the imaginary, Bachelard's theory of knowledge may also be a key to Merleau-Ponty's oscillations and the way he connects metaphors, ontology and an optimistic mood. In his works devoted to applied rationalism, Bachelard argues that phenomena are also constructed: he takes issue with the tendency of contemporary phenomenology—by contrast, with Husserl's phenomenology—to give primacy to receptivity, sensing and an alleged *original given* (Bachelard 1965, p. 2). Phenomenology, or rather what Bachelard wants to call *phenomeno-technique* (Bachelard 1949, pp. 3, 108, 131), should acknowledge the creativity of researchers in the invention of scientific theories and experimental techniques as an integral part of phenomena. "Each new idea remains attached to a perspective of acquisition, an *approximation-structure* [*une structure-approche*] that develops in a sort of *space-time* of essences" (Bachelard 1949, p. 50).[8] The reference to an original intuition or truth is abandoned when one considers the very procedure of approximate knowledge, knowledge essentially consisting in approximation ("*connaissance approchée*"), as ontological at heart. In his philosophy of the imaginary Bachelard comes closer to an existential and ontological formulation of these ideas: "contemplation offers but a superficial view ( . . . ) it keeps us from actively understanding the universe. Action, in its prolonged forms,

imparts more important lessons than contemplation. More particularly a philosophy of opposition [*du contre*] has one step up on the philosophy of unity [*du vers*] for ultimately it is opposition that defines human happiness" (Bachelard 2002, p. 46). "The world is my provocation" (Bachelard [1942] 1993, p. 214). This is actually also an idea that germinates in Merleau-Ponty's work, although in the form of a more understated micro-dynamogeny, for Merleau-Ponty is justifiably suspicious of a too blatant praise of human genius and will. Thus, Merleau-Ponty does combine a philosophy of invigorating opposition [*du contre*] with a philosophy of 'toward' [*du vers*].[9]

In *The structure of behavior*, already, Merleau-Ponty intentionally uses this strategy of successively going down a path, backtracking and entering a contrasting path. In the fourth part of *The structure of behavior*, Merleau-Ponty overplays an idealist version of his theory before asking, in a provocative move, "Is there not a truth of naturalism?". This is what I have described as a "tricky plot" (Dufourcq 2021, §23), a hybrid of literary and philosophical devices. It is, I contend, fruitful to see some of Merleau-Ponty's later texts as implementing a similar strategy. The passage of *The visible and the invisible* in which Merleau-Ponty ventures several schematizations that rectify each other and explicitly call them metaphors perfectly exemplifies such a strategy: "We say therefore that our body is a being of two leaves [*feuillets*] ( ... ) One should not even say, as we did a moment ago, that the body is made up of two leaves ( ... ) If one wants metaphors, it would be better to say that the body sensed and the body sentient are as the obverse and the reverse ( ... ) There is reciprocal insertion and intertwining of one in the other. *Or rather* ( ... ) two circles, or two vortexes" (Merleau-Ponty 1968, pp. 137–38).

It is also illuminating to read Merleau-Ponty's famous working note about metaphors (Merleau-Ponty 1968, pp. 221–22) in light of this link with a micro-dynamogeny. Merleau-Ponty tries to make sense of the phrase "A 'direction' of thought". He claims that defining this phrase as a metaphor "is too much or too little". If thought lends itself to a spatial description, "metaphor" is too little: there is rather a thorough unity. If thought is irreducible to the realm of material objects, even the beginning of a movement of transposition fails to have an ontological value. Should we then renounce this concept of metaphor? Merleau-Ponty's working note leaves us in the middle of the ford. However, Merleau-Ponty also specifies that one "does not have to choose between them [the mind as being nowhere and as being attached]". Therefore, I contend that, when Merleau-Ponty strongly asserts "there is no metaphor" (Merleau-Ponty 1968, pp. 221–22), he is using provocation in a Bachelardian sense: in fact, he wants to keep the two poles without weakening them. In Merleau-Ponty's view, the invisible is really invisible *and* lends itself to transposition. Merleau-Ponty creates a plot to dynamize and dialecticize the circle into a vortex. A "direction" of thought *is a non-metaphor by excess and by default*. This is the complex that it "is". Similarly, "the flesh of the world" is *a non-metaphor by excess and by default.* This definition is, in fact, the last word of Merleau-Ponty's ontology, the same last word that he utters in *Eye and Mind*: "What, says the understanding, like [Stendhal's] Lamiel, *is that all there is to it*? Is this the highest point of reason, to realize that the soil beneath our feet is shifting, to pompously call "interrogation" what is only a persistent state of stupor, to call "research" or "quest" what is only trudging in a circle, to call "Being" that which never fully *is*?" (Merleau-Ponty 1993, p. 149) However, Merleau-Ponty adds, this disappointment issues from a "spurious fantasy [*faux imaginaire*]" that dreams of completion and pure presence and thus fails to enjoy the ultimate truth and thrust of circling in metaphors.

## 7. Conclusions

Merleau-Ponty's later philosophy is unfinished and the oscillations it contains could be regarded as the symptom of a premature and awkward attempt to hold together two incompatible views: a romantic holistic intuition and the traditional existentialist recognition of the dramatic dimension of existence. I do not think that it would be fair to content oneself with this interpretation. Happy metaphors in Merleau-Ponty's work are certainly not reducible to the naïve reference to perfect fullness. The flux of phenomena and emotions

cannot but be manifold. The ultimate ground is, ironically, the imaginary texture of beings. My flesh and the flesh of the world are never fully one; however, both are transposable into the other and, through these transpositions, their intrinsic *dynamic and narrative* being can unfold. It is thus crucial to hold on to the notion of the flesh of the world as being equally and with the same intensity a metaphor and an ontological concept. As such, this figure can evade a gnostic version of existentialism. Indeed, this metaphor returns us to *concrete* interactions between our body, the world, things and other living being. A strength of Merleau-Ponty philosophy is his idea of "indirect ontology" (Merleau-Ponty 1968, p. 179): no direct access to a mysterious Being separated from beings, but rather an ontology that is immanent in a process of reinforcing affective relations with other beings. Here, the fantastic in philosophy takes on the form of magical realism rather than horror. The concept of the flesh of the world can inspire literature, poetry, art, or even biology. In that way, Merleau-Ponty outlines an ontological story that we can take up. His works certainly show all the symptoms of the Chandos complex, but they may also very well be taking up this complex in a more active and intentional way, thus turning it into a method that resonates with Bachelard's philosophy of no. It is actually the destiny of the Chandos complex: a staged separation from the world cannot but be another avatar of the story of being and bear the flesh of the world as its reverse side. Whether intentionally or not, it eventually always begets holistic joyful images.

**Funding:** This research received no external funding.

**Conflicts of Interest:** The author declares no conflict of interest.

## Notes

1. See also (Jonas 1963, pp. 320–40). Gnosticism, a religious and philosophical movement that developed in the first and second centuries CE. Gnostics draw a radical distinction between the good god and the demiurge who is responsible for the disastrous creation of the material world. The latter is regarded as evil and our goal should be to seek salvation. Gnosticism thematizes separation and dualities. First, within the divine realm, second between these human beings who bear a spark of divinity and the world in which they were thrown but that is alien to the genuine divine. Gnostics (from the Greek *gnosis,* knowledge) claim to know clearly what many others are doomed to ignore: only the souls of the chosen ones have this direct and perfect connection to God, that makes them strangers in the material world. As argued by Plotinus in *Against the Gnostics*, this binary model fosters elitism, absolute contempt for this world and a definition of salvation that does not involve a continuous process starting in the world (Plotinus 1952). Heidegger firmly rejects any connection between his philosophy of the Fall (*Verfallenheit*) and Gnosticism. There are indeed many differences between them, above all the definition of *Verfallenheit* as an irreducible dimension of existence instead of a catastrophic event. However, a tendency to jargonize and ossify the difference between authentic and inauthentic (Adorno 1964; Dufourcq 2021, §14), the emphasis consistently placed on estrangement, Angst and the radical difference between Dasein and other beings are elements that lead several thinkers to discern a secret gnostic pattern in Heidegger's philosophy (Jonas 1963; Brisart 1990).

2. "Un trou dans l'être". I have modified the translation: "a penetration in being" (Merleau-Ponty 1960, p. 126).

3. See the concept of *musical ideas* in (Merleau-Ponty 1968, p. 149; [1964] 2001, p. 193).

4. Merleau-Ponty specifies that the identity is that of structures, not a "real" identity, in other words not the identity of substances. This point will be crucial in my approach to Merleau-Ponty's ontology and the Chandos problem: if being is ultimately diacritical, the very concepts of identity and difference become ironical. See also (Merleau-Ponty 1968, p. 261): "In what sense it is *the same* who is seer and visible: the same not in the sense of ideality nor of real identity. The same in the structural sense: same inner framework, same *Gestalthafte,* the same in the sense of openness of another dimension of the 'same' being".

5. See Merleau-Ponty (2003, p. 226): «The human body as symbolism».

6. «Imaginary» here does not mean «fantasized by my imagination»: the imaginary realm is an ontological dimension of beings prior to being the object of my imagination. I have commented at length upon this working note in *Merleau-Ponty: une ontologie de l'imaginaire* (Dufourcq 2012, pp. 359–61).

7. Emmanuel de Saint Aubert also uses this term to define Merleau-Ponty's concept of chiasm in *Le scénario cartésien. Recherches sur la formation et la cohérence de l'intention philosophique de Merleau-Ponty* (de Saint Aubert 2005, p. 170).

8. My translation. "Chaque idée nouvelle reste attachée une perspective d'acquisition, une structure-approche qui se développe dans une sorte d'espace-temps des essences".

9. This is also the key to Merleau-Ponty's hyper-dialectic, as well as to the theme of centrifugal-centripetal vortexes and to the ontological concept of the stabilized explosion (1968, p. 268). This approach is also akin with the theory of schematism in Kant,

with the Husserlian concept of imaginative variations (in a non-idealist form: no absolute intuition of essences, or even essences that could ever be grasped in a perfectly self-coincident intuition) and even with Bergson's taste for dynamic schemas and images (but unlike Bergson, Bachelard and Merleau-Ponty do not see these images and schemas as a means of guiding the reader discursively to the threshold of intuition, an intuition that could only be achieved silently).

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
