# Peer review of "Happy Existentialist Metaphors: Merleau-Ponty’s Flesh of the World and the Chandos Complex"

_humanities, doi:10.3390/h11010017_

Round 1
Reviewer 1 Report
The author develops the idea that the concept of the flesh of the world is introduced by Merleau-Ponty "in an anti-gnostic move to overcome the idea of a fundamental rift between humans and the world". Really no encounter with a real world would ensue from such a radically detached being. Perception must be prepared and motivated from within the perceptible objects, it must stem from an intrinsically phenomenal world, the author writes. This is about the complicity, the dialogue between my flesh and the flesh of the world. The author draws attention to the contradiction in the reasoning of Merleau-Ponty. If the flesh is specifically the flesh of my human body, how can it be the flesh of the world? If the flesh of my body is homogeneous to the matter of the world, the transposition is possible but "its explanatory power vanishes, for there is no specific characteristic that can be transposed in an illuminating manner from my flesh to the flesh of the world." To overcome this contradiction, according to the author, Merleau-Ponty puts the emphasis ambiguously on the “same stuff” the body and the world are made of. Since Merleau-Ponty continues to hope for a conciliation of the opposites author called it the Chandos complex. The author finds similarities between Chandos’ ideas and the key features of Existentialism. The Chandos complex consists in a certain dynamic that orients and gives momentum to our imaginative forces and to our existential relation to the world, the author writes. The contradiction is partially resolved through the concept of "imaginary body". The imaginary body is the institution of being in that there is indeed a fluid transposition of styles of being or diacritical melodies through various beings, including my body. My flesh and the flesh of the world resonate with each other. Therefore, they can’t coincide. It is precisely the differences between styles of being, including, among others, the particularity of my body, that create the space for this resonance to operate. Based on this, the author claims that the only way to think the relation between my flesh and the flesh of things is to define it as a circular relation of mutual institution. "Error" Merleau-Ponty, according to the author, is the approximation of this idea by formulas of identity, sameness, literalness and homogeneity. The author concludes that the notion of the flesh of the world is being equally and with the same intensity a metaphor and an ontological concept. As such, this figure can evade a gnostic version of existentialism. Indeed, this metaphor returns us to concrete interactions between our body, the world, things and other living being. The author's definition of "a gnostic version of existentialism" raises some doubts. It is necessary to give a clearer definition of this in order to distinguish between other concepts of existentialism, which are very close to the ideas of Merleau-Ponty and are not in opposition with them (for example, K. Jaspers or G. Marcel).
Author Response
Many thanks for your review. It's great to see that the main ideas and the purpose of the article are perfectly accurately summarized in your report. Indeed the reference to "a Gnostic version of existentialism" was very elliptical and thus pretty obscure: I totally agree that the article can be improved in this regard, thank you for this suggestion! I have clarified this in the revised version of the article (see attachment note 1): hopefully this will make this intricate reference to Gnosticism more understandable for the reader. Actually, I don't think that existentialism is Gnostic in itself, but what I had in mind is Jonas' reading of Heidegger as well as Merleau-Ponty's allusion to Heidegger's Existentialism as "a gnosis". Moreover Merleau-Ponty knew Plotinus' philosophy very well and the famous "Against the Gnostics" provides interesting arguments against a certain tendency of Existentialism. In fact defining Heidegger's philosophy or Sartre's philosophy as Gnostic is somehow a provocation, I have also clarified that in the footnote: "Gnosticism" here describes some aspects of their philosophy, that actually enter in tension with other ideas they put forward. But at the end of the day, I do want to claim that this gloomy version of existentialism is reviving some deep, ancient Gnostic tendencies namely elitism combined with the myth of an estrangement of humans in the world.
I have also polished the text, see in particular:
- p. 4, line 170--"The tricky task of integrating them ... "
- p. 5, l. 190-191: "arraigns objective thought and warns us against a cultural regimen . . ."
- p. 5, l. 231: I replaced "facetiae" (apparently not a much-used word) with jest
- p. 6, l. 247: "verging on sublimity and farce:
- p. 7, l. 329-330: "we commonly ascribe to things and persons a certain recognizable set of appearances, etc."
- p. 8, lines 353-354: " that can be ascribed neither to things nor to a clear-cut organism."

Reviewer 2 Report
The general thesis put forward in the article is interesting. It requires, however, a deepening in light of most recent secondary literature dealing with the issues discussed in the text.
Author Response
Many thanks for your review. As far as the most recent secondary literature is concerned, the article replies mainly to Malabou and Barbaras. I totally agree that the article does not propose an exhaustive state of the art on this subject, and I will without any doubt unfold this systematic analysis of the debate around Chandos and the myth of a lost union with nature in my forthcoming works (this is one of the first articles I wrote on this topic). The editor of the special issue in which the article will be published agreed with this choice.
Reviewer 3 Report
I had trouble figuring out what this article is about. Literature/literary fiction seems to play an important role. So that should be mentioned as a keyword and in the introduction/abstract. In fact, Chandos seems to be fundamental (in the strict sense of the term) to the paper. It might be a better idea to start with the (rewritten) section on Chandos and the ‘Chandos complex’. And to state in the Introduction that the originality of the research lies in shedding light on the conceptualisation and significance of the ‘flesh of the world’ in the work of Merleau-Ponty by having recourse to Hofmannsthal and what the author has coined as the 'Chandos complex'. The section on Chandos must be properly introduced by starting with the writer Hofmannsthal (and not with a fictional character!) and more clearly rewritten.
The author should try to transcend the level of detailed and specialist discussion to indicate the broader philosophical picture/context/debate in which their research participates. They should be more explicit about the nature of the contribution of their paper to that same debate. Very confusing introduction that is a specialist discussion rather than an introduction. The reader simply does not know what the article is about. What is the main research question? What are the stakes?
I am aware that mystifying language is quite inevitable when dealing with existentialism/phenomenology. However, I deem it the task of the author to ensure that the reader does not get lost in the maze of woolly words. This means bringing structure to the article and not worsening things by using terms/combinations of words that are utterly illogical (such as ‘ontological mood’).
Careful re-reading and editing, and careful choice of words are recommended. Shorter sentences are less confusing. An example of a sentence that must rephrased: ‘This article contends that bright metaphors and magic realism are at least as fundamental and that ontology must come to terms with “the Chandos complex”, namely a form of ambivalence and oscillation between Gnosticism and holism that makes both positions fake and hollow’: the terms ‘fake’ and ‘hollow’ are strong adjectives that are not the most logical with regard to Gnosticism and holism. Besides, it is unclear what the ‘that’ refers to.
The article needs a better, more logical and less chaotic abstract. Start with the problem/background, and then articulate the main argument, while avoiding too many names and discussions. The author may also say something about their approach in the abstract. They should not assume that the reader is acquainted with their discussion and very specific terminology coined by the author (such as the Chandos complex).
It may help to re-structure the paper by beginning each section with the conclusion of the previous section and the aim of the present section, as well as the line and argument and approach in that same section. It may also help to imagine a reader who knows nothing about the topic. How would the author then tell what they wish to tell? And what is the most important insight that the reader would gain by reading the paper?
I think that the originality and merit of the paper will be high if the author succeeds in re-writing a well-structured and well-argued paper. Good luck!
Author Response
Many thanks for your review. I am sorry that the article was so perplexing and confusing. It's an invited contribution to a special issue and, with the agreement of the editor of the said issue, it's designed for specialists in Phenomenology and Existentialism. But I have modified the abstract following your advice. I have also trimmed too long sentences and replaced "ontological mood" with "the mood that is the most attuned to the ontological perspective ".

Round 2
Reviewer 3 Report
The abstract is a great improvement, and the article is now sufficiently referenced. I would have preferred to see a more systematic approach and style, but that’s partly a matter of taste. Moreover, this would most probably have required a non-phenomenological approach to Merleau-Ponty. It is quite challenging to distance oneself from the author whom we deal with. On a side note, I wonder whether the suggested similarity between presocratic science/philosophy and Merleau-Ponty holds water. Congratulations on the forthcoming publication of your article!